# Influence of Butyrate on Impaired Gene Expression in Colon from Patients with High Blood Pressure

**DOI:** 10.3390/ijms24032650

**Published:** 2023-01-31

**Authors:** Jing Li, Elaine M. Richards, Eileen M. Handberg, Carl J. Pepine, Eyad Alakrad, Chris E. Forsmark, Mohan K. Raizada

**Affiliations:** 1Key Laboratory of Plant Germplasm Enhancement and Specialty Agriculture, Wuhan Botanical Garden, Chinese Academy of Sciences, Wuhan 430074, China; 2Department of Physiology and Aging, University of Florida College of Medicine, Gainesville, FL 32610, USA; 3Department of Medicine, Divisions of Cardiovascular Medicine, University of Florida College of Medicine, Gainesville, FL 32610, USA; 4Gastroenterology, Hepatology and Nutrition, Department of Medicine, University of Florida College of Medicine, Gainesville, FL 32610, USA

**Keywords:** hypertension, colonic transcriptome, drug–gene interaction, gut organoid, butyrate

## Abstract

Hypertension (HTN) is associated with gut dysbiosis and the depletion of butyrate-producing bacteria in animal models and people. Furthermore, fecal material transfer from donor hypertensive patients increases blood pressure in normotensive recipient animals and ameliorates HTN-associated pathophysiology. These observations have implications in the impaired interactions between the gut and gut microbiota in HTN. Although this concept is supported in animal models, little is known about human HTN. Therefore, our objective for this study was to compare gene expression with transcriptomics and its potential to influence microbiota in subjects with normal and high blood pressure (HBP). Colon samples from reference subjects with normal blood pressure (REF) and HBP were used for RNA-seq to analyze their transcriptomes. We observed the significant downregulation of gene sets governing immune responses (e.g., *SGK1* and *OASL*), gut epithelial function (e.g., *KRT20* and *SLC9A3R1*), gut microbiota (e.g., *PPARG* and *CIDEC*) and genes associated with cardiovascular and gut diseases (e.g., *PLAUR* and *NLN*) in HBP subjects; the expression of genes within these pathways correlated with blood pressure. Potential drug targets in the gut epithelium were identified using the Drug Gene International Database for possible use in HTN. They include peroxisome proliferator-activated receptor gamma (*PPRG),* active serum/glucocorticoid regulated kinase 1 *(SGK1)* and 3 beta-hydroxysteroid isomerase type II inhibitor *(HSD3B).* Finally, butyrate, a microbiota-derived short-chain fatty acid, restored the disrupted expression of certain functional genes in colonic organoids from HBP subjects. Patients with HBP exhibit a unique transcriptome that could underlie impaired gut–microbiota interactions. Targeting these interactions could provide a promising new therapeutic intervention for hypertension management.

## 1. Introduction

Evidence supports the hypothesis that impaired bidirectional communication between the gut microbiome and gut epithelium leading to gut–brain axis dysfunction is critical in the development and establishment of hypertension (HTN). HTN-associated gut dysbiosis, gut wall pathology and leakiness corroborate this concept [1,2,3,4,5,6]. Additionally, fecal material transfer (FMT) from donor hypertensive patients or animals increases blood pressure (BP) and establishes HTN pathophysiology in recipient animals with normal BP [7,8]. FMT recapitulates all aspects of gut–brain axis dysfunction in HTN. These observations contribute to an emerging concept that gut mucosal–microbiota interactions are important in BP control and alterations in this communication are central to the development and establishment of HTN [2,5,6].

The gut epithelium confers protection from the harmful effects of microbial metabolites and other microbiota-derived molecules, inflammatory cytokines, toxins and immune cells by maintaining a tight gut barrier and immune function [9]. Alterations in epithelial function could compromise gut wall permeability resulting in the leakage of adverse molecules and aberrant afferent signaling with deleterious effects on the cardiovascular system [4,10]. This view is supported by the following: (i) Gut leakiness was demonstrated in HTN in both animal models and humans establishing a critical step for harmful sequelae [3,4,10]. (ii) Changes in certain bacterial metabolites (e.g., lipopolysaccharide, butyrate, tryptophan metabolites) have been observed in HTN [10,11,12]. Supplementation with butyrate or butyrate-producing bacteria reduces BP [4,13,14,15] and a clinical trial is underway to test this in humans (ClinicalTrials.gov Identifier: NCT04415333). (iii) Transcriptomics have shown distinct gene expression profiles in colon and colonic organoids from spontaneously hypertensive rats (SHR) compared with normotensive Wistar Kyoto (WKY) rats. This includes altered expression in pathways regulating T-cell receptor signaling, immunity, antigen presentation in MHC class I and barrier function [16,17]. Together, these observations suggest that epigenetic changes in the SHR gut epithelium may influence microbiota and, in turn, BP.

Despite evidence for the involvement of epithelial cells in animal models of HTN, little is currently known about their role in gut–microbiota communication in human HTN. It is crucial to fill this gap in knowledge to appreciate translational benefits and develop epigenetics-based targets for HTN. Furthermore, animal models do not completely recapitulate human HTN and there are significant differences in the gut microbiota of rodents and humans. Therefore, our overall objective for this study was to compare colonic transcriptomes of reference subjects with normal BP (REF) and high BP (HBP) towards this goal. The colon was used because it plays a key role in the water and salt balance, harbors the largest population of bacteria compared with other gut regions [11,13,18,19], is the site of active bacterial metabolism, and shows dysbiosis, gut wall pathology and leakiness in HTN [3,10,16]. Furthermore, the colonic epithelium and 3D organoid cultures of the colon respond to signals from the environment and microbial metabolites [16,17,20]. We evaluated the hypotheses that HBP subjects exhibit altered gene expression in the colonic epithelium relevant to immunity, antigen presentation, epithelial proliferation, differentiation and signaling, and that butyrate regulates these genes.

## 2. Results

### 2.1. Characterization of Subjects and Colonic Tissues

HBP and REF subjects did not differ in characteristics other than BP (Table 1). Systolic blood pressure was used as the dependent variable for confounder analyses against the independent clinical variables. Although chronic kidney disease was a confounding factor, there was no significant regression between BP and chronic kidney disease (*p* = 0.06) when adding other characteristics. Thus, no independent variables affected BP in our cohort. PCA analysis of the transcriptomes of all samples revealed no outliers (Figure 1A). In addition, the expression of phenotypic markers of cell populations of the colon were comparable in REF and HBP groups in both the colonic epithelium (Figure 1B, *p* > 0.05) and organoids (Appendix A). These markers included ALPI (enterocytes), MUC2 (goblet cells), CHGA (enteroendocrine cells), LGR5 (CBR stem cells), LZY (Paneth cells) and BMI1 (+4 cells: secretory and enteroendocrine progenitors, +4 stem cells, Paneth cell precursors/label-retaining cells). Together, this indicates the consistency of colon biopsies and sequencing among subjects and groups, thus all samples were carried forward into downstream analyses.

### 2.2. HBP Patients Have Distinct Colon Transcriptome Signatures

Transcriptomic profiling of colonic biopsies from REF and HBP revealed thirty-five genes that were differentially regulated between the two groups; 6 down and 29 upregulated (FDR-corrected *p* < 0.05, absolute fold change ≥ 1.5) in the HBP compared with REF subjects (Figure 2A). Enrichment analysis (GSEA) of differentially expressed genes showed that gene sets associated with gut immune responses and immunity were underrepresented in HBP colon transcriptomes, including isoprenoid biosynthesis, antigen processing and presentation in MHC class I, defense responses to virus and neutrophil activation in immune responses (Figure 2B). This suggests inefficient immune responses in the gut epithelium of subjects with HBP. For example, decreased antigen presentation of MHC class I (normalized enrichment score (NES) = −1.8082, adjusted *p* = 0.0076) and defense responses to virus (NES = −1.4717, adjusted *p* = 0.014). APOBEC1, OASL and BIRC3 may prevent mounting of effective immune defense against gut dysbiosis and gut luminal contents in HBP.

Isoprenoid biosynthesis was significantly downregulated in the colonic epithelium of HBP subjects (Figure 2B, NES = −1.9108, adjusted *p* = 0.013). Isoprenoids play an important role in the genesis and development of cardiovascular disorders [21], such as cardiac hypertrophy and fibrosis, endothelial dysfunction and fibrotic responses of smooth muscle cells, and may have relevance here as we have previously reported increased gut fibrosis in animal models of hypertension [3]. Interestingly, viral gene expression (NES = 2.1693, adjusted *p* = 0.00084) including viral transcription, processing of mRNA and viral translation (http://geneontology.org, accessed on 28 July 2021) was upregulated. Conversely, pathways such as histone modification and epigenetic regulation of gene expression were over-represented, suggesting that these may be involved in altered transcriptomic pathways. We found similar differential gene sets in colonic organoids, including antigen presentation, viral transcription, SRP-dependent co-translational protein targeting and nuclear-transcribed mRNA catabolic process (Figure 3).

The specific genes altered between REF and HBP subjects in selected pathways are illustrated in heat maps (Figure 2C). mRNAs for immune response (e.g., PDLIM2, STOM, BIRC3 and OASL) were decreased in HBP subjects (Figure 2C). Notably, mRNAs for most genes associated with cardiovascular diseases (e.g., HSD3B2, PPARG, NLN and CEACAM1) were downregulated in the colons of HBP subjects (Figure 2C). For example, polymorphisms of HSD3B2 (hydroxy-delta-5-steroid dehydrogenase, 3 beta- and steroid delta-isomerase 2), a crucial enzyme for steroid synthesis, is associated with aldosterone levels and BP [22]. In contrast, MYADM (myeloid-associated differentiation marker) was increased in HBP subjects (Figure 2C). Furthermore, the mRNAs of genes associated with gut diseases were decreased in the HBP subjects (Figure 2C). Examples include: CLCA4 (chloride channel accessory 4), which codes for the calcium-activated chloride channel family and is involved in colorectal cancer [23], and SLC26A3 (solute carrier family 26 member 3), a transporter for chloride/bicarbonate exchange and the maintenance of intestinal epithelial barrier integrity [24]. Finally, other genes whose expression could impact gut epithelium-microbiota dysfunction include those that were significantly decreased in HBP subjects: CIDEC, PPARG, and CEACAM1 are involved in host gene–gut microbiota interactions, while EMP1, KRT20, SLC9A3R1 and UCA1 are essential for gut epithelial differentiation and proliferation (Figure 2C).

Next, we used qPCR to validate the RNA-seq data of key genes. Figure 2D shows that, consistent with the RNA-Seq data, mRNA for *CEACAM1*, *SLC26A3*, *ABCG2*, *CIDEC*, *SGK1*, *KRT20*, *EMP1*, *PLAUR* and *PPARG* were significantly decreased while mRNA for *MYADM* was increased in HBP subjects (Figure 2C).

### 2.3. Identification of Potential Novel Drug Targets

We used the Drug Gene International Database (dgidb.org) to identify any genes with altered expression in the colonic epithelium of the HBP subjects as potential targets of existing drugs. The DGidb annotates genes of interest with respect to known drug–gene interactions based on selected pathways, functions and gene families [25]. Table 2 shows a list of differentially expressed genes that are drug targets. For example, MBX-2044 and Treprostinil are agonists of PPARG that was decreased in the HBP group. These drugs are currently being investigated for the treatment of type 2 diabetes and pulmonary arterial hypertension (ClinicalTrials.gov Identifiers: NCT00422487, NCT03835676).

### 2.4. Specific Differentially Expressed Genes (DEGs) Correlated with SBP and DBP

Pearson correlation analysis was performed between the SBP/DBP and RPKM of DEGs to investigate the DEGs most closely associated with BP. Pixel mapping showed that *KRT20*, *MXI1*, *PDLIM2*, *SLC9A3R1* and *VSTM1* negatively correlated, while *MYADM* positively correlated with BP (Figure 4).

### 2.5. Butyrate Treatment Rescued Impaired Gene Expression in Organoids from HBP Subjects

Butyrate-producing bacteria are depleted in hypertensive animals and patients [1,4]. Furthermore, butyrate supplementation alleviates HTN pathophysiology in a preclinical model [4,13]. These findings led us to determine if butyrate treatment would rescue impaired gene expression in HBP subjects. We modeled this in human colon organoids since testing in human subjects was not possible. Three-dimensional organoids from crypts containing the colonic stem cells of subjects with HBP (Figure 5) were established and treated with 3 mM butyrate for 24 h.

High-throughput RNA sequencing followed by PCA analysis revealed a clear distinction between the transcriptomes of untreated and butyrate-treated organoids from HBP subjects (Figure 6A). We observed 3831 upregulated genes (red dots) and 702 downregulated genes (blue dots) in butyrate-treated organoids (Figure 6B). GSEA of these differentially expressed genes disclosed that gene sets for histone modification were significantly inhibited by butyrate in organoids from HBP subjects (Figure 6C). They included gene sets associated with histone methylation, with a normalized enrichment score (NES = −1.7663, adjusted *p* = 0.0089, e.g., *PRDM5* and *BRCA1*) and histone acetylation (NES = −1.8117, adjusted *p* = 0.002, e.g., *MSL3* and *JADE2*) (Figure 6B). Additionally, the gene set for viral gene expression was downregulated by butyrate in the organoids of HBP subjects (NES = −1.8241, adjusted *p* = 0.00096, e.g., *MDFIC* and *TRIM27*): it was enriched in the colonic tissue of the HBP group compared to the REF group (Figure 2B and Figure 6C). Hormone metabolic process functions as a determinant of both the beneficial and detrimental effects of sex hormones on the cardiovascular system [26]; it was upregulated in the HBP group by butyrate (NES = 1.7658, adjusted *p* = 0.00094, e.g., *IYD* and *RETSAT*) but impaired in the colonic tissue of the HBP group compared to the REF group (Figure 2B and Figure 6C). Furthermore, RPKM data demonstrated that butyrate activated the expression of genes related to epithelial differentiation and proliferation (*KRT20* and *SLC9A3R1*), immune response (*REAG1A*, *DHRS9*, *STOM*, *MXI1*, *BIRC3*, *OASL*, *CEACAM1* and *SGK1*) and gut microbiota (*CIDEC* and *CEACAM1*) in HBP organoids (Figure 7), whose expression was downregulated in the colon tissues of HBP subjects. Of these genes, some were involved in cardiovascular (*ABCG2*, *CEACAM1* and *SGK1*) and gut diseases (*REAG1A*, *ABCG2*, *BIRC3*, *CEACAM1* and *SGK1*), and others were potential drug targets (*ABCG2*, *BIRC3*, *CEACAM1* and *SGK1*) (Figure 7). Taken together, these data suggest that butyrate treatment recues key functional genes in the organoids from HBP subjects. This is consistent with butyrate’s multifunctional effects (such as anti-inflammatory activity, a reduction in oxidative stress, etc.) in the improvement of gut barrier functions.

## 3. Discussion

This study demonstrates the salient differences in gene expression profiles between the epithelia of colons of subjects with REF and HBP supporting the conclusion that impaired gut epithelial–microbiota crosstalk could be associated with high BP in patients. Thus, it validates the developing concept of a link between dysfunctional communication between the host and gut microbiota in human HTN.

We found that the DEG of HBP subjects were predominantly decreased in expression in the colonic epithelium (29/35 DEG were downregulated) while histone modification was a pathway upregulated in their transcriptomes. The mechanism may be gut microbiota dependent. Short chain fatty acids (SCFA)-producing bacteria have been shown to be depleted in HTN. SCFA play a crucial role in gut epithelial cells, including as inhibitors of histone deacetylases. Th epigenetic regulation of gene expression is partially dependent upon the histone modification activity of histone deacetylases. The depletion of SCFA increases histone deacetylation that, in turn, generally decreases gene transcription. This suggests the regulation of human colonic epithelium gene expression by the gut microbiota.

We have previously demonstrated that colonic antigen processing and presentation (AP) in MHC class I was diminished in a rodent model of HTN [17]. MHC class I antigen processing and presentation in the colonic epithelium was similarly decreased in human HBP subjects, supporting and extending our previous findings. Changes in AP, particularly MHC class I, could be important in HTN as it transmits molecular signals generated by the gut microbiota to epithelial cells to regulate proliferation and differentiation and, thus, influences gut, peripheral- and neuro-inflammatory status.

Two genes involved in defense responses to viruses are pertinent: *OASL* (oligo adenylate synthase-like protein) and *BIRC3* (baculoviral IAP repeat containing 3). *OASL* is interferon-inducible, degrades viral RNA and activates pattern recognition receptors [27], while *BIRC3* encodes apoptosis inhibitors, is regulated by many cytokines and protects against certain viruses [28]. Decreases in both are likely to impair HBP subjects’ abilities to mount effective antiviral responses. Two genes involved in isoprenoid biosynthesis are downregulated in HBP subjects, potentially impacting immune modulation and the handling of retinoic acid (RA) in the gut epithelium. *AIDH1A3* (aldehyde dehydrogenase 1 family member A3) encodes aldehyde dehydrogenase, a key enzyme in RA formation, and is highly expressed in intestinal epithelial cells [29]. *DHRS9* (dehydrogenase/reductase 9) encodes dehydrogenase/reductase 9, abundant in the human colon and involved in RA metabolism [29,30]. The isoprenoid signaling pathway has been implicated in CVD [21]. Finally, we found increased viral gene expression, the relevance of which remains unknown. The possibility that HBP subjects exhibit some degree of viral infection in the colonic cells cannot be ruled out at present. This is consistent with our shotgun metagenomic data (unpublished) showing that ~55% of fecal samples from HBP subjects contained viruses not present in samples from REF subjects, including *Papillomaviridae* and *Alloherpesviridae*.

It is important to discuss several other genes because they are targets of established drugs and important in epithelial–microbiota communication. *PPARG* (peroxisome proliferator activator receptor gamma), a ligand-activated transcription factor affecting fatty acid oxidation and lipid metabolism, is implicated in metabolic disease (e.g., obesity and diabetes) [31,32]. Additionally, PPARG’s protective role in HTN has been known for some time [33]. Colonic PPARG may exert epigenetic influences via transactivation by SCFAs. This is supportive of evidence of well documented decreases in butyrate-producing bacteria in HTN [3,4,16,19]. Furthermore, its interactions and activation by many bacterial taxa and species including *Roseburia* and other *Firmicutes* gives *PPARG* unique properties to modify gut bacterial communities [34,35]. We found PPARG is a key gene associated with gut diseases and immune response pathways in addition to HTN, CVD and microbiota (Figure 2). A recent study disclosed that the knock-out of *PPARG* enhanced the management of BP via the regulation of the RAS system in hypertensive rats [36]. Together with evidence of decreases in butyrate-producing and other interactive bacteria, it is reasonable to suggest that colonic PPARG could be a druggable target for the treatment of HTN, particularly considering its highly significant interactions with Mbx-2044 and Treprostinil.

*SGK1* and *HSD3B1* are relevant in steroidogenesis, mineralocorticoid receptor signaling and BP control. *SGK1* plays a role in the regulation of certain ion channels impacting the epithelial (including colonic) excretion of Na+, and influences renal inflammation, fibrosis and BP [37], whereas polymorphisms of *HSD3B1* are linked with BP [38]. Another gene, *ABCG2*, an ABC transporter, is important in transcellular barriers and regulates the absorption of nutrients, drugs and orally ingested toxins [39]. Its activity produces beneficial effects; thus, it could be a useful drug target. *CEACAM1* is in the immunoglobulin superfamily, functioning as an MHC class 1-independent receptor on natural killer T-cells and binding bacteria and viruses to inhibit specific immune responses [40]. Decreased *CEACAM1* expression is likely to compromise immune responses. Finally, *PLAUR*, a plasminogen activator receptor is an interesting candidate for consideration because of the well-established role of the plasminogen activator-plasminogen activator inhibitor system in HTN and CVD [41,42]. Together, these observations point out that genes regulating immune status, mineralocorticoid metabolism, and key transporters could be potential targets for repurposing drugs to effect BP and control HTN.

Three other genes, although not on the drug target list, are worthy of consideration because of their role in the maintenance of overall epithelial function. *KRT20* (keratin 20) is a major type 1 cytokeratin in the mature enterocytes and goblet cells involved in mucin production [43]. Decreased expression of *KRT20* would have a major impact on epithelial integrity and supports our evidence of decreased goblet cells in animal models of HTN [3]. MYADM (myeloid-associated differentiation marker) organizes membrane rafts that interact with MUC-1 to influence mucin [44]. It is positively correlated with HTN and hypoxia-induced pulmonary hypertension involving TGF-beta signaling [45,46,47]. The increased expression of *MYADM* in this study is consistent with its known role in HTN.

Our study, for the first time, used colon organoids from subjects with HBP to show that butyrate treatment rescued altered expression of genes associated with immune modulation, such as the suppression of nuclear factor kappa B activation and the inhibition of histone deacetylase. For example, *CEACAM1* was upregulated (Figure 7). CEACAM1 belongs to a carcinoembryonic antigen family functioning as an MHC class I-independent inhibitory receptor. Its binding to bacterial and viral adhesins impairs immune-cell function and inhibits intestinal inflammation [48]. Interestingly, *ceacam1* null deletion led to the elevation of BP and renal dysfunction in mice [49]. Thus, an increase in *CEACAM1* expression by butyrate would be beneficial to attenuate HTN-associated gut inflammation. The enrichment of butyrate-producing bacteria in WKY rats verses SHR has been correlated with the increased expression of gene sets such as antigen presentation, immune response and cell junction organization in the gut epithelium [6,17]. This study, showing that butyrate treatment increased the expression of genes for immune response that have depressed expression in organoids from HBP subjects, is consistent with observations in hypertensive animals. Supplementation of prebiotic fiber or the reintroduction of SCFAs to fiber-depleted mice had protective effects on the development of HTN, cardiac hypertrophy and fibrosis via the cognate SCFA receptors GPR43/GPR109A [13]. Together, these data suggest that supplementation by butyrate or butyrate-producing bacteria attenuates HTN and restores gut health in animal models, although the mechanism remains to be elucidated [4,14,15]. Although an association between SCFAs and HTN is strong in animal models, data from human HTN are mixed [4,50,51,52] and the discrepancy remains to be reconciled with further studies. Finally, it is pertinent to point out that we found significantly higher numbers of butyrate-responsive genes in organoids of HBP subjects compared with differences observed in the colonic epithelium between HBP and REF subjects. This difference remains to be reconciled. It may be due to an enhanced accessibility and greater sensitivity to butyrate in our ex vivo organoid cultures, or the lack of competition from gut microbiota to metabolize butyrate in the organoid culture setting. However, variations due to the small number of subjects donating colonic tissues for analysis cannot be discounted at the present time.

## 4. Materials and Methods

### 4.1. Study Cohort

Medical records of subjects scheduled for routine screening colonoscopy were reviewed. Subjects were approached if they were 18–80 years old, weighed more than 110 lbs and had either high BP or normal BP at previous office visits using the 2017 ACC/AHA definition for HTN. Exclusion criteria included: a history of autoimmune disease or other chronic inflammatory conditions, antibiotic treatment within two months of study enrollment, current use of medications known to modify gut microbiota, probiotic use, or history of intestinal surgery. Subjects meeting these criteria and who were competent and willing to provide written consent were recruited. We received the consent of 17 subjects with high SBP (151 ± 4 mmHg) and 18 subjects with normal BP (122 ± 2 mmHg) for the study. Blood pressures were measured again on the day of the procedure with automated BP devices after subjects were supine for at least 5 min. Based on this reading, we excluded one subject who no longer exhibited high blood pressure (blood pressures for this group were 156 ± 5 mmHg, HBP, n = 16) while all 18 subjects recruited for normal blood pressure were retained and an additional subject was identified with normal BP on the day of the procedure (110 ± 4 mmHg, REF, n = 19). During colonoscopy, colonic biopsies (~2 × 4 mm) from descending aspects of the colon were collected from these subjects and used in the study. To identify clinical characteristics as confounding factors affecting BP, linear regression was performed using SPSS software version 28.

### 4.2. RNA-seq

Within 30 min of the collection of biopsies, they were rinsed 3 times with cold PBS, preserved in RNA and stored at −80 °C until further analysis. Total RNA was extracted from colonic biopsies or colonic organoids with a RNeasy Plus Mini Kit (Qiagen, Hilden, Germany). Total RNA with 28S/18S >1 and RNA integrity numbers 7.1–9.3 were used for RNA-Seq library construction. cDNA was generated using a SMART-Seq HT kit (Takara Bio, Otsu, Japan) and RNA-Seq libraries were constructed with a Nextera DNA Flex Library Prep kit and Nextera DNA Unique Dual Indexes Set A (Illumina, California, USA.) and sequenced on the NovaSeq6000 (Illumina) at the University of Florida NextGen DNA Sequencing Core Facility (at 2 × 150 bp, aiming for 50 million paired-end reads per sample). The DNA fragment size distribution of the library averaged 600 bp (range 150–1500 bp) (TapeStation, Agilent, California, USA.). Quantitative PCR (qPCR) assays were performed for the library quantification (2 nM starting concentration and final loading concentration of 300 pM). RNA seq analysis was performed using a CLC genomics workbench (Qiagen 21.0.4, Hilden, Germany.). Data were normalized to reads per kilobase of transcript per million mapped reads (RPKM). False Discovery Rate (FDR)-corrected *p* value < 0.05 was used as the criterion for significance. We used principal component analysis (PCA) to classify samples and evaluate outliers. PCA, based on singular value decomposition, was performed on transcriptomes of colonic tissues using the prcomp package in R statistical software.

Gene set enrichment analysis (GSEA) was performed using iDEP V0.94 (South Dakota, USA) and the pre-ranked fgsea method [53]. Heat maps of differentially expressed genes (DEGs) in enriched pathways were built using supervised hierarchical clustering in Heatmapper [54].

### 4.3. Quantitative PCR (qPCR)

To validate the RNA-seq findings in the gut epithelium, the cDNAs used for RNA-seq were also used to perform qPCR (ABI Prism 7600, Foster City, CA, USA) with Taqman universal PCR master mix and specific probes (*CEACAM1*: Hs05041713_s1, *SLC26A3*: Hs00995363_m1, *ABCG2*: Hs01053790_m1, *CIDEC*: Hs01032998_m1, *SGK1*: Hs00178612_m1, *KRT20*: Hs00300643_m1, *MYADM*: Hs01880197_s1, *EMP1*: Hs00995363_m1, *PLAUR*: Hs00958880_m1 and *PPARG*: Hs01115513_m1). *GAPDH* (Hs02758991_g1) was used to normalize the expression of these genes and fold change was calculated relative to the REF group using the−ΔΔCt method.

### 4.4. Human Colonic 3D Organoid Culture

Primary colonic crypts were isolated from descending aspects of the colons of REF and HBP subjects with gentle cell dissociation reagents (STEMCELL Technologies) including 2 mM EDTA for 90 min. They were grown and maintained as 3D-spheroid cultures in Matrigel (BD Biosciences) containing organoid growth medium (STEMCELL Technologies) with recombinant human Noggin [PeproTech] and EGF [BioLegend], recombinant human IGF-1, FGF-basic [FGF-2; BioLegend] and R-spondin1 [R&D], Y-27632 [STEMCELL Technologies], and A83-01 [Tocris], as described previously [17,20,55].

Colonic organoids were cultured for 8 days, then treated with 3.0 mM butyrate (Sigma-Aldrich, St. Louis, MO, USA) for 24 h in the following experiments. Selection of the butyrate dose was based on our published data [17], demonstrating that a 3.0 mM dose was optimal for gene expression without affecting organoid growth [20]. This is consistent with doses of butyrate in colonic epithelial cells (2 mM) in regulation of the assembly of tight junctions [56], in Caco-2 cells (10 mM) for apoptosis [57] and its levels in the gut, portal, hepatic and venous blood [58].

### 4.5. Correlation Analysis

Correlation analyses were performed on BP data and the RPKM of DEGs. Pearson correlation coefficients (r) were generated for all 35 samples regardless of group, using MATLAB software with a cutoff value of 0.48 (N = 16, *p* < 0.05) and displayed using pixel maps [6].

## 5. Conclusions

This study underscores the importance of data from human subjects to preclinical studies since differences exist between animal models of HTN and human HTN. The gut microbiota of rodents and humans [59] differ significantly and HTN-associated shifts in gut bacterial populations are distinct [1,3,4,7]. Transcriptional profiling of colonic epithelial cells showed focused differences in gene profiles in humans compared with SHR [16,17]. Although similar trends in alterations in biological pathways were seen, the genes involved were distinct (Figure 2 and Figure 3) [17]. This suggests that bacterial and gene targets for HTN therapies would differ in SHR and human HBP subjects.

It is increasingly evident that gut dysbiosis and a dysfunctional gut–brain axis plays important roles in the development and establishment of hypertension. Our study disclosed that colonic genes relevant to gut immunity, epithelial proliferation, differentiation and signaling are altered in subjects with high BP. This could, in part, be rescued by butyrate treatment. This provides possible mechanistic information on dysfunctional gut–microbiome communication in human hypertension and potential genes for testing for hypertension management.

## Figures and Tables

**Figure 1 ijms-24-02650-f001:**
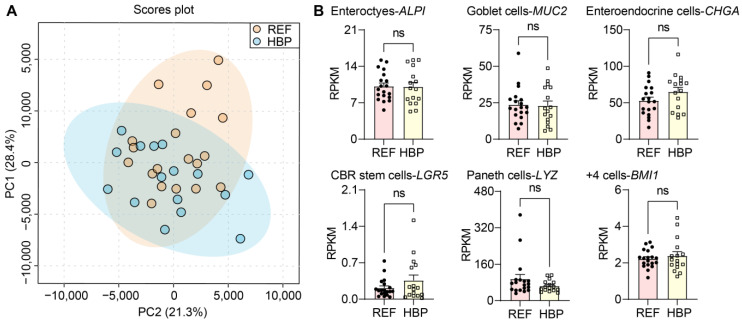
Quality control of transcriptomic methods and colon sampling. Transcriptome profiling. (**A**) Principal component analysis (PCA) plot based on complete expression profiles. Blue region is for 95% confidence interval of REF group, yellow region is for 95% confidence interval of HBP group, and blue and yellow dots are representative of the samples of REF and HBP group labelling with systolic blood pressure. (**B**) Expression of markers of cell type in colon. Reads per kilobase per million (RPKM) values are means ± standard error of the mean (SEM). ns *p* > 0.05, unpaired *t* test or Mann–Whitney U test.

**Figure 2 ijms-24-02650-f002:**
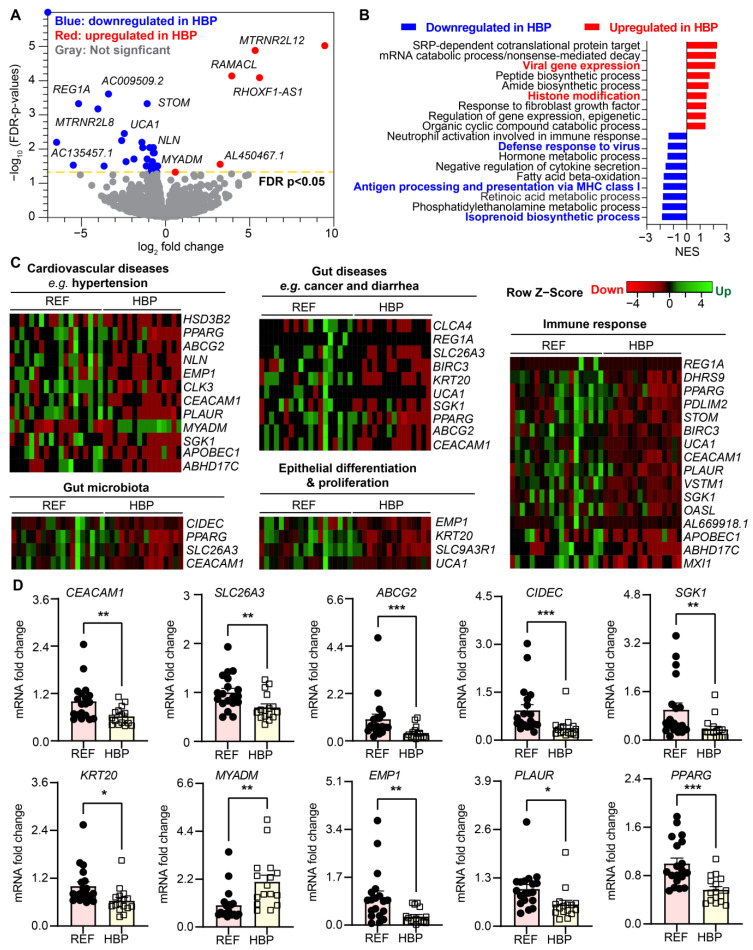
Comparison of the transcriptomes of colons of REF and HBP subjects. (**A**) Volcano plot showing differentially expressed genes (DEGs) in colonic epithelium from REF (n = 19) with HBP (n = 16) subjects. FDR-corrected *p* < 0.05, blue circles, downregulated DEGs; red circles, upregulated DEGs in HBP subjects; gray circles, genes with no difference in expression. (**B**) Pathways significantly enriched in colon of HBP subjects compared with control subjects (FDR-corrected *p* < 0.05). NES: normalized enrichment score. Negative NES (blue bar) shows downregulated pathways and positive NES (red bar) shows upregulated pathways in HBP subjects. (**C**) Supervised heat maps of genes in the enriched gene sets. Enriched gene sets were determined by gene set enrichment analyses performed using supervised hierarchical clustering in Heatmapper with FDR-corrected *p* < 0.05, upregulated (red) gene and downregulated (green) gene in HBP vs. REF subjects. (**D**) mRNA expression of representative genes in colonic tissues from HBP and REF subjects by qPCR. Fold changes relative to REF group. GADPH expression was used for normalization. Values are means ± SEM, * *p* < 0.05, ** *p* < 0.01, *** *p* < 0.001, unpaired *t* test or Mann–Whitney U test.

**Figure 3 ijms-24-02650-f003:**
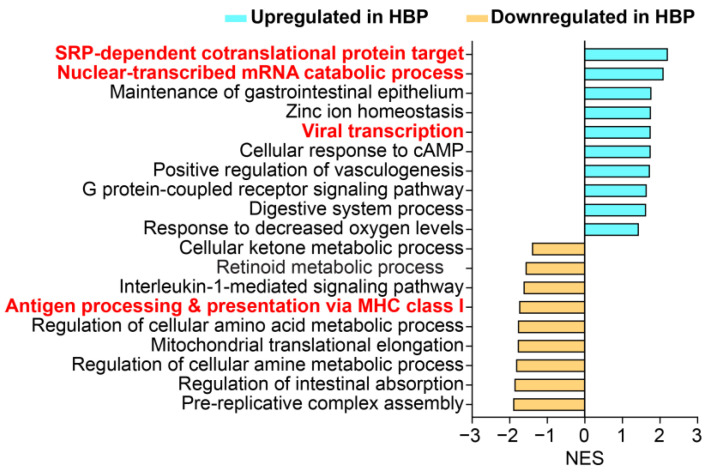
Pathways significantly enriched in colonic organoids from HBP subjects compared with REF subjects (FDR-corrected *p* < 0.05). NES: normalized enrichment score. Negative NES (Light blue bar) shows downregulated pathways and positive NES (Yellow bar) shows upregulated pathways in colonic organoids from HBP subject.

**Figure 4 ijms-24-02650-f004:**
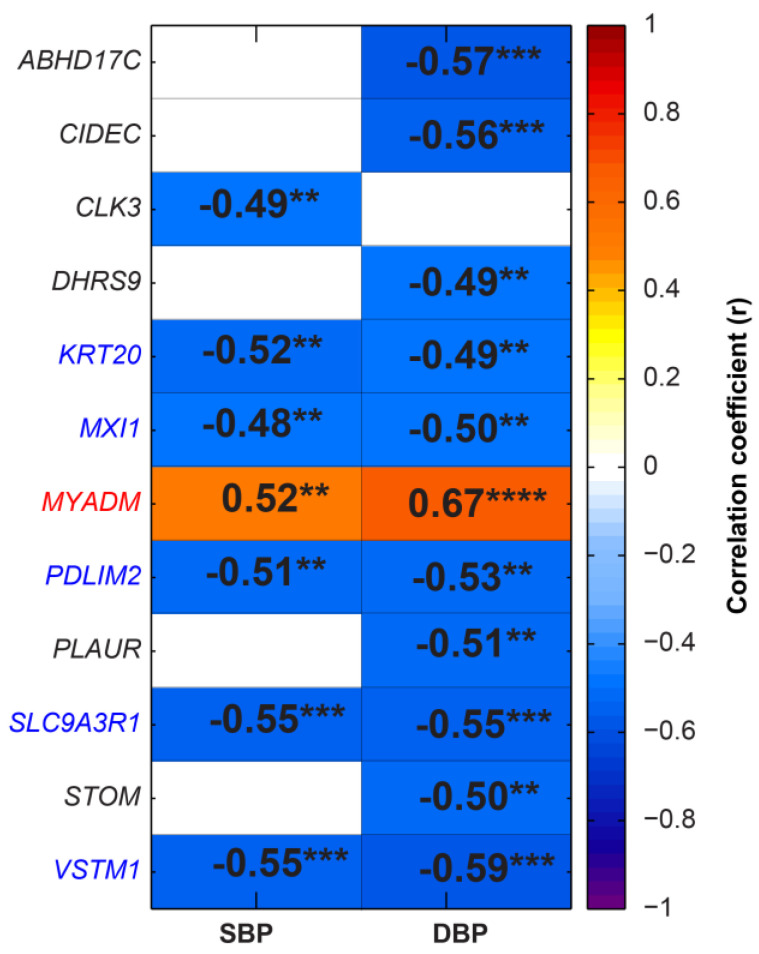
Correlation analysis between blood pressure and expression of the DEGs in REF (n = 19) and HBP subjects (n = 16). Pixel map representation of the correlation between blood pressure and expression of the DEGs in REF and HBP subjects. Positive r indicates the positive correlation and negative r indicates the negative correlation. (** *p* < 0.01, *** *p* < 0.001 and **** *p* < 0.0001, cutoff value |r| is 0.48).

**Figure 5 ijms-24-02650-f005:**
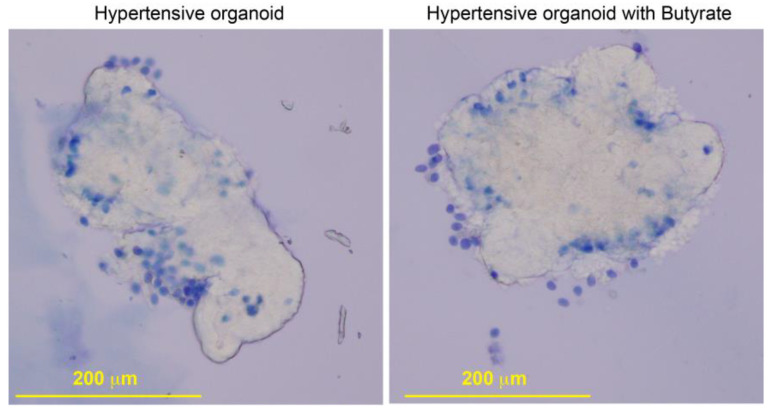
Phase microscopic images documenting growth of organoids from isolated colonic crypts of hypertensive subjects. Organoids were stained with trypan blue; dead cells are blue. Scale bar: 200 μm.

**Figure 6 ijms-24-02650-f006:**
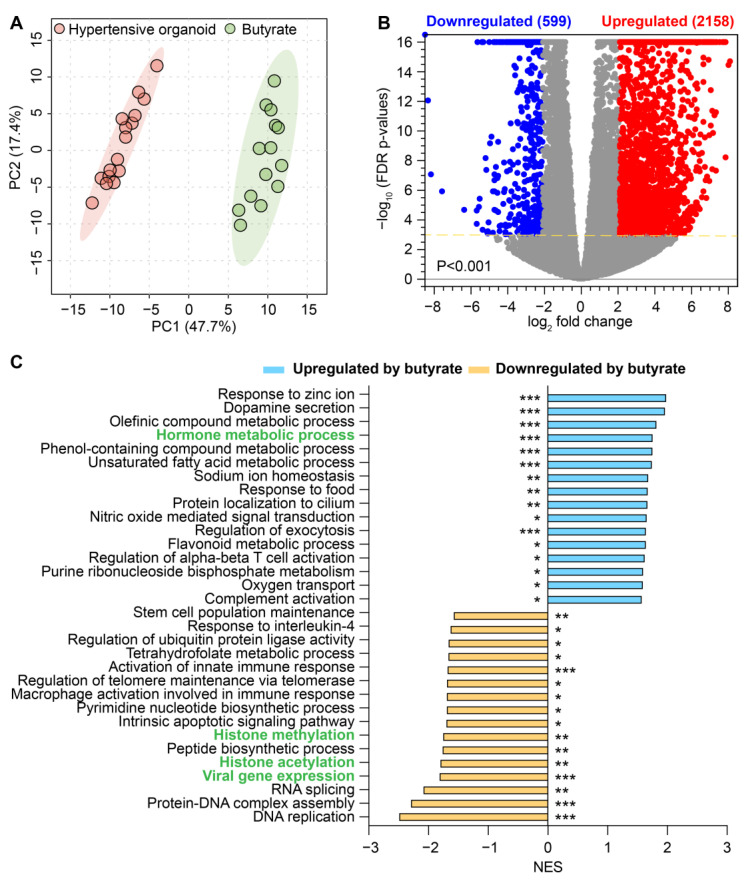
Comparison of the transcriptomes of colonic organoids without and with butyrate in HBP subjects. (**A**) Principal component analysis (PCA) plot based on complete expression profiles. Light red region is for 95% confidence interval of human organoids without butyrate (n = 14) and light green region is for 95% confidence interval of human organoids with butyrate (n = 16) in HBP subjects. (**B**) Volcano plot showing differentially expressed genes (DEGs) in human organoids without and with butyrate in HBP subjects. FDR-corrected *p* < 0.001, absolute fold change ≥ 4. Blue circles, downregulated DEGs; red circles, upregulated DEGs in human organoids with butyrate; gray circles, genes with no difference in expression. (**C**) Pathways significantly enriched in human organoids with butyrate compared to without butyrate treatment. Negative NES (blue bar) shows downregulated pathways and positive NES (yellow bar) shows upregulated pathways in human organoids with butyrate. Green-highlighted pathways were dysregulated in HBP colonic tissue, whose pathways could be restored by the butyrate treatment. FDR-corrected * *p* < 0.05, ** *p* < 0.01, *** *p* < 0.001.

**Figure 7 ijms-24-02650-f007:**
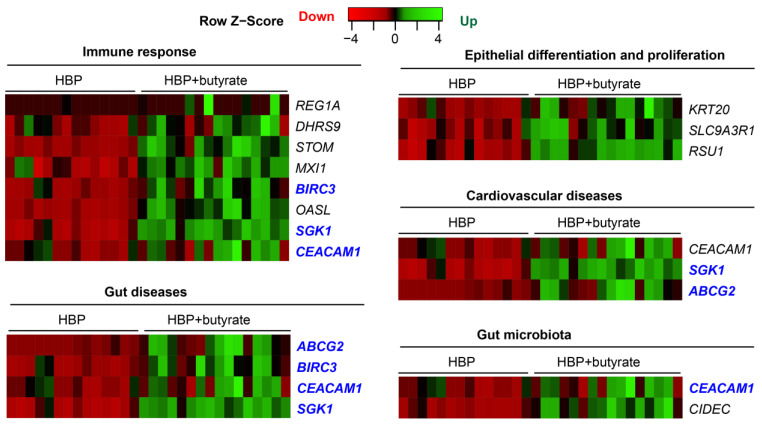
Butyrate rescues impaired gene expression of colonic organoids from HBP subjects with the annotated pathways or drug target information. Differentially expressed genes were shown using supervised hierarchical clustering in Heatmapper with FDR-corrected *p* < 0.05, upregulated (red) gene and downregulated (green) gene in HBP organoids vs. HBP organoids treated with butyrate. Blue-highlighted genes were the potential drug targets.

**Table 1 ijms-24-02650-t001:** Characteristics of subjects donating colon biopsies for study. *p* values were calculated using unpaired *t*-tests or contingency tests (Fisher’s exact) where appropriate.

**Variable**	**RBP-Mean (95% CI)**	**HBP-Mean (95% CI)**	***p* Value**
Age (years)	56 (52–60)	61 (57–65)	0.07
Men	6 (32)	8 (50)	0.32
Systolic blood pressure (mmHg)	111 (104–118)	156 (146–165)	<0.0001
Diastolic blood pressure (mmHg)	68 (62–73)	88 (84–91)	<0.0001
**Variable**	**RBP-Number (%)**	**HBP-Number (%)**	***p* Value**
White Race	15 (79)	10 (62.5)	0.45
Body mass index (kg/m^2^)	29.0 (27–31)	30.1 (28–33)	0.32
Smoker	4 (21)	4 (25)	>0.99
Diabetes	3 (16)	4 (25)	0.68
Hyperlipidemia	5 (26)	1 (6)	0.19
Coronary artery disease	0	0	>0.99
Peripheral Artery disease	0	0	>0.99
Cerebrovascular disease	1 (5)	0	>0.99
Chronic kidney disease	2 (11)	0	0.49
**Number of Anti-Hypertensive Drug**	**RBP-Number (%)**	**HBP-Number (%)**	***p* Value**
0	12 (63)	8 (50)	0.30
1	6 (32)	5 (31)	>0.99
2	1 (5)	2 (12)	0.58
3	0	1 (6)	0.46
≥4	0	0	>0.99
**Type of Anti-Hypertensive Drug**	**RBP-Number (%)**	**HBP-Number (%)**	***p* Value**
RAS blockers (ACEIs or ARBs)	2 (11)	6 (37)	0.11
Aldosterone antagonists	0	0	>0.99
Beta blockers	1 (5)	2 (12)	0.58
Calcium channel blockers	3 (16)	3 (19)	>0.99
Diuretics	2 (11)	1 (6)	>0.99
Alpha blockers	0	0	>0.99
Arteriole vasodilators	0	0	>0.99

**Table 2 ijms-24-02650-t002:** Representative list of drug–gene interactions of HBP-associated DEGs to identify potential novel drug targets in the colon. The Drug Gene Interaction Database (https://www.dgidb.org/search_interactions, accessed on 28 July 2021) was used to identify potential therapeutic targets in the colon for hypertension management and control.

Drug or Antibody	Gene	Fold Change	FDR P	Mechanism of Interaction
Mbx-2044	*PPARG*	–1.43	0.0486	Peroxisome proliferator-activated
Treprostinil				receptor gamma modulator
Aldosterone	*SGK1*	–2.65	0.0067	Activate serum/glucocorticoid
				regulated kinase 1
AT-406	*BIRC3*	–1.85	0.0094	cIAP1/cIAP2 inhibitor
Trilostane	*HSD3B2*	–3.55	0.0199	3-beta-hydroxysteroid dehydrogenase/
				isomerase type II inhibitor
Arcitumomab	*CEACAM1*	–1.57	0.0327	Antigen monoclonal antibody
Urokinase	*PLAUR*	–1.66	0.0319	Plasminogen activator
Chembl488910	*ABCG2*	–1.78	0.0431	Resistance protein inhibitors

## Data Availability

RNA seq data (normalized transcript expression, per million mapped reads, RPKM) have been deposited at https://docs.google.com/spreadsheets/d/1Ji99W41QgKhwt6F3XwXZD4xh6-Hud25S/edit?usp=sharing&ouid=113702272944965355681&rtpof=true&sd=true. The data were accessed on 1 August 2021.

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
