# Peer review of "Influence of Butyrate on Impaired Gene Expression in Colon from Patients with High Blood Pressure"

_ijms, 2023, doi:10.3390/ijms24032650_

Round 1
Reviewer 1 Report
This manuscript by Jing Li and colleagues explores the transcriptional profiles of the colonic epithelial cells or colon organoids from hypertension patients compared with regular blood pressure cohort. They highlighted
The topic of High Blood Pressure and gut microbiota is extremely interesting. The incorporation of human patient colon samples and organoids makes this study results more clinically relevant and convincing. There are some small points in the paper that need to be further improved.
(1) the conclusion part needs to revise. In the current version, it seems more like a discussion session instead of summarizing the findings and significance in this manuscript.
(2) for figure 7, it is not straightforward and hard to interpret when the authors use bar graphs with different symbols to show involved pathways. I would recommend the authors should group the genes by the pathways they are in. Or use a heatmap to show the pathway clustering and RNA changes.
(3) line 66, “because it plays a key a role in water and salt balance” delete the second “a”.
Author Response
Reviewer 1
This manuscript by Jing Li and colleagues explores the transcriptional profiles of the colonic epithelial cells or colon organoids from hypertension patients compared with regular blood pressure cohort. They highlighted The topic of High Blood Pressure and gut microbiota is extremely interesting. The incorporation of human patient colon samples and organoids makes this study results more clinically relevant and convincing. There are some small points in the paper that need to be further improved.
Response:
We thank the reviewer for encouraging comments.
(1) the conclusion part needs to revise. In the current version, it seems more like a discussion session instead of summarizing the findings and significance in this manuscript.
Response:
We have revised conclusion part to summarize the findings and significance in this manuscript.
“It is increasingly evident that gut dysbiosis and a dysfunctional gut-brain axis play important roles in development and establishment of hypertension. Our study disclosed that colonic genes relevant to gut immunity, epithelial proliferation, differentiation and signaling are altered in subjects with high BP. This could in part, be rescued by butyrate treatment. This provides a possible mechanistic information on dysfunctional gut-microbiome communication in human hypertension and potential genes for testing for hypertension management.”
(2) for figure 7, it is not straightforward and hard to interpret when the authors use bar graphs with different symbols to show involved pathways. I would recommend the authors should group the genes by the pathways they are in. Or use a heatmap to show the pathway clustering and RNA changes.
Response:
We have revised Figure 7 that shows heatmap as suggested.
(3) line 66, “because it plays a key a role in water and salt balance” delete the second “a”.
Response:
We have deleted this repeating word “a” in the manuscript.
Reviewer 2 Report
This manuscript is an exciting and complete analysis of colon tissue in hypertensive individuals. It is well-written, easy to read, and has important information for basic as well as clinical research. There are some points however which will enhance the manuscript. There should be a table stating the age, gender and time in which hypertension was diagnosed and the time of the treatment. This information is critical for the readers as is table 1. Also, the abstract should be improved since it is general and not attractive and it does not represent the amount of data the authors have been able to obtain. There is also a lack of discussion on the genes that were encountered up and downregulated in hypertensive patients as compared to data available from colon cancer patients. The effect of butyric acid is interesting; however, the authors should discuss better the relevance to the clinic and its relevance in cardiovascular diseases.
In general a good manuscript that requires small modifications.
Author Response
Reviewer 2
This manuscript is an exciting and complete analysis of colon tissue in hypertensive individuals. It is well-written, easy to read, and has important information for basic as well as clinical research. There are some points however which will enhance the manuscript.
Response:
Thanks for your comments.
There should be a table stating the age, gender and time in which hypertension was diagnosed and the time of the treatment. This information is critical for the readers as is table 1.
Response:
We have revised “Table 1” stating the age and gender. Our approved IRB protocol (no. 201903360) did not include determination of when subjects were diagnosed with HTN. Thus, this data was not collected/available.
Also, the abstract should be improved since it is general and not attractive and it does not represent the amount of data the authors have been able to obtain.
Response:
We have revised the abstract.
There is also a lack of discussion on the genes that were encountered up and downregulated in hypertensive patients as compared to data available from colon cancer patients.
Response:
We have added the discussion on the genes that were encountered up and downregulated in hypertensive patients.
“A recent study disclosed that knock-out of PPARG enhanced the management of BP via the regulation of RAS system in hypertensive rats35.
- Yuan, J.;Wang, L.; Han, S.; Wang, Z.; Ni, Y.; Geng, Y.; Zhang, L., PPARG Silencing Improves Blood Pressure Control and Alleviates Renal Damage by Modulating RAS Circadian Rhythm in Hypertensive Rats. Annals of Clinical & Laboratory Science 2022, 52 (3), 452-461.
Interestingly, ceacam1 null deletion led to the elevation of BP and renal dysfunction in mice48.
- Li, C.;Culver, S. A.; Quadri, S.; Ledford, K. L.; Al-Share, Q. Y.; Ghadieh, H. E.; Najjar, S. M.; Siragy, H. M., High-fat diet amplifies renal renin angiotensin system expression, blood pressure elevation, and renal dysfunction caused by Ceacam1 null deletion. American Journal of Physiology-Endocrinology and Metabolism 2015, 309 (9), E802-E810.”
The effect of butyric acid is interesting; however, the authors should discuss better the relevance to the clinic and its relevance in cardiovascular diseases.
Response:
We have been very careful not to overinterpret our data because of mixed results on butyrate in human hypertension. However, we have added the following statement in response to the comment:
Although an association between SCFAs and HTN is strong in animal models, data from human HTN is mixed4, 49-51 and the discrepancy remains to be reconciled with further studies.
- Kim, S.;Goel, R.; Kumar, A.; Qi, Y.; Lobaton, G.; Hosaka, K.; Mohammed, M.; Handberg, E. M.; Richards, E. M.; Pepine, C. J., Imbalance of gut microbiome and intestinal epithelial barrier dysfunction in patients with high blood pressure. Clinical science 2018, 132 (6), 701-718.
- De la Cuesta-Zuluaga, J.;Mueller, N. T.; Álvarez-Quintero, R.; Velásquez-Mejía, E. P.; Sierra, J. A.; Corrales-Agudelo, V.; Carmona, J. A.; Abad, J. M.; Escobar, J. S., Higher fecal short-chain fatty acid levels are associated with gut microbiome dysbiosis, obesity, hypertension and cardiometabolic disease risk factors. Nutrients 2018, 11 (1), 51.
- Huart, J.;Leenders, J.; Taminiau, B.; Descy, J.; Saint-Remy, A.; Daube, G.; Krzesinski, J.-M.; Melin, P.; De Tullio, P.; Jouret, F., Gut microbiota and fecal levels of short-chain fatty acids differ upon 24-hour blood pressure levels in men. Hypertension 2019, 74 (4), 1005-1013.
- Tilves, C.;Yeh, H. C.; Maruthur, N.; Juraschek, S. P.; Miller, E.; White, K.; Appel, L. J.; Mueller, N. T., Increases in circulating and fecal butyrate are associated with reduced blood pressure and hypertension: results From the SPIRIT trial. Journal of the American Heart Association 2022, 11 (13), e024763.
In general a good manuscript that requires small modifications.
Response:
Thanks, all suggested modifications have been made in the revised manuscript.